# Redefining enteroaggregative *Escherichia coli* (EAEC): Genomic characterization of epidemiological EAEC strains

Nadia Boisen[1]*, Mark T. Østerlund[1], Katrine G. Joensen[1], Araceli E. Santiago[2], Inacio Mandomando[3], Alejandro Cravioto[4], Marie A. Chattaway[5], Laura A. Gonyar[2], Søren Overballe-Petersen[1], O. Colin Stine[6], David A. Rasko[7], Flemming Scheutz[1], James P. Nataro[2]

**1** Statens Serum Institut, Department of Bacteria, Parasites and Fungi, Copenhagen, Denmark, **2** University of Virginia School of Medicine, Department of Pediatrics, Charlottesville, Virginia, United States of America, **3** Centro de Investigação em Saúde da Manhiça (CISM), Maputo, Mozambique, **4** Universidad Nacional Autónoma de México, Faculty of Medicine, Mexico City, Mexico, **5** Public Health England, Gastrointestinal Bacteria Reference Unit (GBRU), Colindale, United Kingdom, **6** University of Maryland School of Medicine, Department of Epidemiology and Public Health, Baltimore, Maryland, United States of America, **7** University of Maryland School of Medicine, Institute for Genome Sciences, Department of Microbiology and Immunology, Baltimore, Maryland, United States of America

* nbo@ssi.dk

**Data Availability Statement:** All relevant data are within the manuscript and its Supporting Information files.

## Abstract

Although enteroaggregative *E. coli* (EAEC) has been implicated as a common cause of diarrhea in multiple settings, neither its essential genomic nature nor its role as an enteric pathogen are fully understood. The current definition of this pathotype requires demonstration of cellular adherence; a working molecular definition encompasses *E. coli* which do not harbor the heat-stable or heat-labile toxins of enterotoxigenic *E. coli* (ETEC) and harbor the genes *aaiC*, *aggR*, and/or *aatA*. In an effort to improve the definition of this pathotype, we report the most definitive characterization of the pan-genome of EAEC to date, applying comparative genomics and functional characterization on a collection of 97 EAEC strains isolated in the course of a multicenter case-control diarrhea study (Global Enteric Multi-Center Study, GEMS). Genomic analysis revealed that the EAEC strains mapped to all phylogenomic groups of *E. coli*. Circa 70% of strains harbored one of the five described AAF variants; there were no additional AAF variants identified, and strains that lacked an identifiable AAF generally did not have an otherwise complete AggR regulon. An exception was strains that harbored an ETEC colonization factor (CF) CS22, like AAF a member of the chaperone-usher family of adhesins, but not phylogenetically related to the AAF family. Of all genes scored, *sepA* yielded the strongest association with diarrhea ($P = 0.002$) followed by the increased serum survival gene, *iss* ($p = 0.026$), and the outer membrane protease gene *ompT* ($p = 0.046$). Notably, the EAEC genomes harbored several genes characteristically associated with other *E. coli* pathotypes. Our data suggest that a molecular definition of EAEC could comprise *E. coli* strains harboring AggR and a complete AAF(I-V) or CS22 gene cluster. Further, it is possible that strains meeting this definition could be both enteric bacteria and urinary/systemic pathogens.

**Funding:** This study was supported by DFF1333-00156 to N.B.O and The National Institutes of Health grants AI-033096 and AI-125181 to J. P. N. The funders had no role in study design, data collection and analysis, decision to publish, or preparation of the manuscript.

**Competing interests:** The authors have declared that no competing interests exist.

## Author summary

Enteroaggregative *E. coli* (EAEC) has been implicated as a common cause of diarrhea in multiple settings and associated with linear growth faltering among children in low-income countries. Unlike other *E. coli* pathotypes EAEC stands alone in employing a phenotypic, rather than genotypic, definition. Therefore, the lack of a formally recognized genetic definition for EAEC serves to complicate its epidemiologic investigation. In an effort to improve the definition of this pathotype, we generated the most definitive characterization of the pan-genome of EAEC by performing whole genome sequencing on a collection of strains isolated from different geographic settings. We identified a genetic signature of EAEC which could comprise of *E. coli* strains harboring the EAEC transcriptional activator and its adhesin dependent factors. Further, we found that EAEC strains harbor additional putative virulence genes previously reported in extraintestinal pathogenic *E. coli* (ExPEC) and, therefore, strains meeting the re-definition could be both enteric bacteria and urinary/systemic pathogens.

## Introduction

Enteroaggregative *E. coli* (EAEC) has been implicated as a common cause of acute and persistent diarrhea [1,2] and as a cause of linear growth faltering among children in low-income countries [3]. The German outbreak of Shiga toxin (Stx)–producing O104:H4 EAEC highlights the epidemiologic and genomic plasticity of this pathotype [4]. Since its initial description, EAEC has been defined as *E. coli* that do not express the enterotoxins of enterotoxigenic *E. coli* (ETEC), and which adhere to HEp-2 epithelial cells in an aggregative (AA) phenotype [5]. As such, EAEC stands alone among the DEC groups in employing a phenotypic, rather than genotypic, definition. Unfortunately, as nearly all DEC detection is currently accomplished using gene detection, the lack of a formally recognized genetic definition for EAEC serves to impede its epidemiologic investigation.

EAEC pathogenesis is thought to comprise mucosal inflammation and epithelial damage [6], and strains express a heterogeneous array of putative virulence factors [7–11] encoded on the bacterial chromosome or the EAEC-specific pAA plasmid. Most EAEC strains harbor a transcriptional activator of the AraC/XylS class called AggR [12], which controls the expression of plasmid-borne and chromosomal genes; among the AggR-dependent factors are five known variants (designated I-V) of the aggregative adherence fimbriae (AAFs) [13–18]. AAF mediate adherence to human mucosal sections and thereby, presumably, promote intestinal colonization [19]. AAFs may directly incite both cytokine release and the opening of epithelial tight junctions [20,21]. AggR activates expression of its own repressor, called Aar (AggR-activated regulator) [22]. AggR also activates expression of the genes encoding dispersin (*aap*), the dispersin translocator Aat, and the Aai chromosomal type 6 secretion system (*aaiA-Y*) [23]. EAEC strains often harbor a variable number of serine protease autotransporters of *Enterobacteriaceae* (SPATEs) [10], which are implicated in immune evasion, mucosal damage, secretogenicity, and colonization [24]. Notably, none of these AggR-dependent or AggR-independent factors are found in all EAEC isolates and, moreover, no single factor has ever been consistently implicated epidemiologically in EAEC diarrhea [7,8,25]. Strains harboring the AggR regulon or its constituents have been termed typical EAEC [26], and these strains were the focus of this work. Strains that adhere to HEp-2 cells but do not carry AggR-regulated genes have never been evaluated in human volunteer studies, and are therefore of uncertain pathogenicity.

In the current study, we employed whole genome sequence comparative genomics and functional characterization to examine the diversity of 97 EAEC strains isolated from Bangladesh, Mali or Mozambique in the course of a prospective multicenter case-control diarrhea study (Global Enteric Multi-Center Study, GEMS). Genomes were characterized using novel genome mining algorithms to identify virulence factors and to associate them with clinical outcome.

## Methods

### Ethics statement

This research involved genomic characterization of EAEC isolates obtained from participants in the Global Enteric Multicenter Study (GEMS). The ethical review methods for this study were described in detail [27].

### Study design

Isolates comprised 97 strains (57 cases and 40 controls) from Bangladesh (15 cases and 11 controls), Mali (25 cases and 19 controls), and Mozambique (17 cases and 10 controls). The strains were isolated in the course of the GEMS multicenter case-control study [27]. Briefly, GEMS is a 3-year, prospective, age-stratified, matched case-control study of moderate-to-severe diarrhea in children aged 0–59 months residing in censused populations at four sites in Africa and three in Asia. Children with moderate-to-severe diarrhea seeking care at health centers along with one to three randomly selected matched community control children without diarrhea were recruited. From patients with moderate-to-severe diarrhea and controls, both clinical and epidemiological data were obtained, as well as anthropometric measurements and a fecal sample to identify enteropathogens at time of enrolment. The isolates were identified at the study site as EAEC based on multiplex PCR [7] for *aatA* and *aaiC*. Strains were subsequently characterized by PCR for presence of a known AAF-encoding gene, in order to intentionally over-represent strains that did not possess one of the known AAF variants. The EAEC genomes were compared with a set of previously sequenced archetype *E. coli* strains (S1 Table).

### Adherence to colonoid monolayers

Human colonoid line 75C was established from de-identified biopsy tissue from a healthy subject provided informed consent at Johns Hopkins University. All methods were carried out in accordance with guidelines and regulations approved by the Johns Hopkins University Institutional Review Board (IRB# NA_00038329). Colonoids were maintained as previously described [28]. Colonoid monolayers were cultured in a 96-well plate pre-coated with human collagen IV (30 μg/ml; Sigma-Aldrich, USA) until confluency and then differentiated for 3–4 days. EAEC strains were grown for 16–18 hours without shaking in DMEM-high glucose (Gibco). Approximately 1 x $10^6$ CFUs were added to each well. After a 3 hour incubation, the monolayers were washed three times with PBS, lysed with 1% Triton X-100/PBS, and adherent bacteria were enumerated by dilution and plating on Luria agar.

### HEp2 cell assay

Adherence of *E. coli* strains was assessed using HEp-2 cell monolayers as previously described [5].

### Serotyping

Somatic (O) and flagella (H) antigens were identified as described [29]. In addition to a defined antigen number, several additional designations were included: (1) "O rough," i.e. the

boiled culture auto-agglutinated, suggesting absence of O antigen; (2) "O?," i.e. it could not be determined whether the strain produces an O antigen (precipitation with Cetavlon indicating an acidic polysaccharide that could represent capsular K antigen); and (3) "O+," i.e. the O antigen is present but could not be typed. Serotyping was performed at the International *Escherichia* and *Klebsiella* Centre (World Health Organization), Department of Microbiological Surveillance and Research, Statens Serum Institut, Copenhagen, Denmark.

## Genomic methods

Genomic DNA was purified from the EAEC isolates using the DNeasy Blood and Tissue kit (Qiagen). Sequencing was carried out with the Illumina technology on a MiSeq sequencing machine, using the Nextera XT Library preparation protocol (Illumina) for obtaining paired-end reads of 150 bp or 250 bp. Raw data for all isolates were deposited at the European Nucleotide Archive (ENA).

WGS data were pre-processed employing a QC-pipeline (available at https://github.com/ssi-dk/SerumQC). Sequences were removed in the case of contamination with more than 5% of another genera and sequences representing isolates with genome sizes outside the range of 4.64 Mbp-5.56 Mbp. Sequences were removed from the dataset if assemblies comprised of > 350 contigs. *De novo* assembly was carried out using CLC Genomics Workbench 10.

## Nanopore sequencing of pAA plasmid from strain C671-15

Three µg DNA was purified with a Qiagen-tip 100 Plasmid Midi Kit (#12145) and sequenced as previously described [30]. Base-calling was carried out using with Albacore v2.0.2. For long-read sequencing, plasmid DNA from strain C671-15 was isolated using a Qiagen-tip 100 Plasmid Midi Kit (#12145). Three µg of plasmid DNA was fragmented by ten passes through a size G23 needle. The library was constructed using the 1D Ligation Barcoding Kit (SQK-LSK108 and EXP-NBD103, Oxford Nanopore Technologies, Oxford, UK) with 'End-prep' incubations increased to 2x30 minutes and sequenced in an R9.4 flow cell (FLO-MIN106) with a MinION Mk1B (ONT). ONT's Albacore v2.0.2 was used for base calling. ONT sequencing adaptors were removed with Porechop v0.2.0 [31]. Illumina reads were end-trimmed to q20 using Trimmomatic v0.36 [32]. The Hybrid assembly of Illumina and Nanopore reads, generated on NextSeq sequencing machines and ONT, were performed with Unicycler (v0.4.0) [33]. Using CLC Genomics Workbench (v10.1.1), inspection of BLAST-mapped Nanopore with reads longer than 10,000 bp confirmed the structure of the assembled plasmids. Likewise, CLC mappings of Illumina short-reads were inspected to correct eventual small-scale assembly errors (variants, homopolymers and INDELs).

## Analysis using Center for Genomic Epidemiology mining tools

Assembled sequence data were analyzed using typing tools from the Center for Genomic Epidemiology (CGE) (DTU, Lyngby, Denmark) (https://cge.cbs.dtu.dk/services/). CGE contains databases for *in silico* multilocus sequence typing (MLST), serotyping (SerotypeFinder), virulence analysis (VirulenceFinder), *fim*-typing, and antibiotic resistance prediction (consisting of annotated allelic variants for genes encoding serotype, virulence factors and antimicrobial resistance, https://bitbucket.org/genomicepidemiology/workspace/projects/DB).

VirulenceFinder has just been updated and identifies 139 putative or confirmed virulence genes, of which 75 are DEC-associated, 44 are ExPEC-associated, and 20 are found in almost all *E. coli* strains, irrespective of pathotype; in addition to those genes we included another ~60 virulence and putative virulence genes found in DEC and ExPEC.

The various "finder" tools use a BLAST-based approach to detect the genes of interest in the *de novo* assembled genome, and subsequently identifies them against the appropriate reference database. Detection standard parameters were as follows: for VirulenceFinder and SerotypeFinder, 85% sequence identity and 60% sequence coverage; for ResFinder, 90% sequence identity and 60% sequence coverage; for FimTyper, 95% sequence identity and 60% sequence coverage; for MLSTFinder, using the seven loci (*adk*, *gyrB*, *fumC*, *icd*, *mdh*, *purA*, and *recA*) scheme. Results are listed in S2 Table.

## Phylogenetic analysis

The 97 EAEC genomes analyzed in this study were compared with nine previously sequenced EAEC reference genomes, which include the five AAF fimbrial type reference isolates, JM221, 17–2, C227-11, TY-2482, 042, 55989, C1010-00, 34B, and C338-14, as well as 37 diverse *E. coli* and *Shigella* genomes (S1 Table) using the SNP-based *In Silico* Genotyper (ISG) as previously described [34]. ISG identified 234,371 conserved SNPs that were used to infer a maximum-likelihood phylogeny using RAxML v7.2.8, with the GTR model of nucleotide substitution, the GAMMA model of rate heterogeneity, and 100 bootstrap replicates. The phylogeny was midpoint-rooted and diagramed using the interactive Tree of Life software (iTOL v.3).

We investigated differences in gene content among the EAEC genomes sequenced in this study using large-scale BLAST score ratio (LS-BSR) analysis as previously described [35]; protein-coding reading frames were predicted in each of the EAEC genomes using Prodigal [35]. Genes with a BSR ≥0.8 were considered to be positive. The presence of each gene or region among the EAEC genomes from cases and controls were analyzed for statistical significance using Fisher's exact test.

## Results

### Frequency of AggR regulon-associated genes

To ascertain the distribution of EAEC-associated genes among the 97 isolates, we performed WGS and applied the *E. coli* genome mining algorithms VirulenceFinder 1.3, ResFinder 3.1, and SerotypeFinder 2.0 [36]. VirulenceFinder includes genes commonly found in Shiga-toxin producing *E. coli* (STEC), enteropathogenic *E. coli* (EPEC), *Shigella*/enteroinvasive *E. coli* (EIEC), and EAEC. We also performed BLAST search for virulence genes described in ETEC, diffusely adherent *E. coli* (DAEC), and extraintestinal pathogenic *E. coli* (ExPEC), which have not yet been included in the CGE VirulenceFinder (Table 1). Further, per study protocol, EAEC strains in GEMS possessed the plasmid-borne *aatA* and/or the chromosomal *aaiC* gene, thereby fulfilling the current criterion for typical EAEC strains; the *aatA* gene was found in 75% of strains, and the *aaiC* gene in 38%.

Genome analysis suggested that the EAEC strains were mosaic in nature. The dispersin gene, *aap*, was the most frequently detected among all EAEC (89%), followed by *aggR* (84%), and the genes encoding the proteins ORF4 (76%) and ORF 3 (72%) (Table 1). As shown in Table 1, and previously reported, there was a high degree of concordance between genes known to comprise the AggR regulon [7,14,37,38]; such was not the case for two additional AggR regulated genes, *aar* (58%) and *aaiC* (38%).

### Allelic diversity of Aggregative adherence Fimbria (AAF) pilin-encoding genes

Fundamental to EAEC pathogenesis is colonization of the intestinal mucosa, thought to be mediated by AAF, of which five distinct variants have been described. We found evidence of a

**Table 1. Prevalence and Distribution of Putative Virulence Genes in 97 EAEC Isolates from Cases and Controls by Whole Genome Sequence.**

| Gene | Gene description | Pathotype[a] Species | Case | Control | Total | Odds Ratio |
|------|------------------|---------------------|------|---------|-------|------------|
| | | | N = 57 (%) | N = 40 (%) | N = 97 (%) | |
| aggR | Transcriptional activator | EAEC | 45 (78.9) | 36 (90.0) | 81 (83.5) | 0.4 |
| aar | AggR-activated regulator | EAEC | 27 (47.4) | 29 (72.5) | 56 (57.7) | 0.3 |
| aaiC | AaiC, secreted protein | EAEC | 21 (36.8) | 16 (40) | 37 (38.1) | 0.9 |
| aap | Dispersin, antiaggregation protein | EAEC | 48 (84.2) | 38 (95.0) | 86 (88.7) | 0.3 |
| aatA | Dispersin transporter protein | EAEC | 40 (70.2) | 33 (82.5) | 73 (75.3) | 0.5 |
| aggA | AAF/I fimbrial subunit | EAEC | 11 (19.3) | 11 (27.5) | 22 (22.7) | 0.6 |
| aafA | AAF/II fimbrial subunit | EAEC | 3 (5.3) | 5 (12.5) | 8 (8.2) | 0.4 |
| agg3A | AAF/III fimbrial subunit | EAEC | 3 (5.3) | 6 (15) | 9 (9.3) | 0.3 |
| agg4A | AAF/IV fimbrial subunit | EAEC | 14 (24.6) | 7 (17.5) | 21 (21.6) | 1.3 |
| agg5A | AAF/V fimbrial subunit | EAEC | 7 (12.3) | 9 (22.5) | 16 (16.5) | 0.5 |
| ORF3 | Cryptic protein | EAEC | 37 (64.9) | 33 (82.5) | 70 (72.2) | 0.4 |
| ORF4 | Cryptic protein | EAEC | 41 (71.9) | 33 (82.5) | 74 (76.3) | 0.5 |
| capU | Hexosyltransferase homolog | EAEC | 36 (63.2) | 31 (77.5) | 67 (69.1) | 0.5 |
| air | Enteroaggregative immunoglobulin repeat protein | EAEC | 11 (19.3) | 6 (15) | 17 (17.5) | 1.4 |
| eilA | Salmonella HilA homolog | EAEC | 14 (24.6) | 11 (27.5) | 25 (25.8) | 0.9 |
| anr | AraC Negative Regulators | ETEC | 1 (1.8) | 1 (2.5) | 2 (2.1) | 0.7 |
| sigA | IgA protease-like homolog | EAEC/EIEC | 2 (3.5) | 6 (15) | 8 (8.2) | 0 |
| sepA | *Shigella* extracellular protease | EAEC/EIEC | 27 (47.4) | 7 (17.5) | 34 (35.1) | **4.2**\* |
| sat | Secreted autotransporter toxin | EAEC/ExPEC | 12 (21.1) | 6 (15) | 18 (18.6) | 1.5 |
| pic | Serine protease precursor | EAEC/EIEC | 20 (35.1) | 15 (37.5) | 35 (36.1) | 0.9 |
| pet | Plasmid-encoded toxin | EAEC | 4 (7) | 4 (10) | 8 (8.2) | 0.7 |
| espI | Type III secreted proteins | EPEC | 2 (3.5) | 1 (2.5) | 3 (3.1) | 1.4 |
| astA | Heat-stable enterotoxin 1 | EAEC | 21 (36.8) | 15 (37.5) | 36 (37.1) | 1 |
| celB | Endonuclease colicin E2 | EPEC | 1 (1.8) | 3 (7.5) | 4 (4.1) | 0.2 |
| clpK1 | Clp ATPase and thermal stress survival | *Klebsiella* | 1 (1.8) | 1 (2.5) | 2 (2.1) | 0.7 |
| CS22 | Colonization factors, CS22 | ETEC | 3 (5.3) | 0 (0) | 3 (3.1) | - |
| iha | Adherence protein | STEC | 11 (19.3) | 15 (37.5) | 26 (26.8) | 0.4 |
| ireA | Siderophore receptor | ExPEC | 0 (0) | 1 (2.5) | 1 (1) | 0.0 |
| iss | Increased serum survival | ExPEC/APEC | 27 (47.4) | 10 (25) | 37 (38.1) | **2.7Δ** |
| lpfA | Long polar fimbriae | STEC/EPEC | 27 (47.4) | 20 (50) | 47 (48.5) | 0.9 |
| mchB | Microcin H47 part of colicin H | STEC/EPEC | 10 (17.5) | 13 (32.5) | 23 (23.7) | 0.4 |
| mchC | MchC protein | STEC/EPEC | 10 (17.5) | 13 (32.5) | 23 (23.7) | 0.4 |
| mchF | ABC transporter protein MchF | STEC/EPEC | 11 (19.3) | 12 (30) | 23 (23.7) | 0.6 |
| mcmA | Microcin M part of colicin H | STEC/EPEC | 7 (12.3) | 3 (7.5) | 10 (10.3) | 1.7 |
| nfaE | Diffuse adherence fibrillar adhesin gene | ETEC/DAEC | 0 (0) | 2 (5) | 2 (2.1) | 0.0 |
| papGII | P-fimbriae | ExPEC | 1 (1.8) | 0 (0) | 1 (1) | - |
| papA | P-fimbriae | ExPEC | 1 (1.8) | 0 (0) | 1 (1) | - |
| sfaA | S-fimbriae | ExPEC | 0 (0) | 2 (5) | 2 (2.1) | 0.0 |
| papC | Pilus-associated protein C | ExPEC | 1 (1.8) | 0 (0) | 1 (1) | - |
| chuA | *E. coli* haem-utilization gene | ExPEC | 16 (28.1) | 10 (25) | 26 (26.8) | 1.1 |
| hra | Heat-resistant agglutinin | EAEC/ExPEC | 10 (17.5) | 9 (22.5) | 19 (19.6) | 0.8 |
| iutA | Aerobactin receptor | ExPEC | 5 (8.8) | 3 (7.3) | 8 (8.2) | 1.1 |
| KpsMII | Translocate group II capsular polysaccharides | ExPEC | 23 (40.4) | 12 (30) | 35 (36.1) | 1.6 |
| kpsE | Translocate group II capsular polysaccharides | ExPEC | 23 (40.4) | 12 (30) | 35 (36.1) | 1.6 |
| fyuA | Yersiniabactin receptor | ExPEC | 37 (64.9) | 19 (47.5) | 56 (57.7) | 2.0 |

*(Continued)*

**Table 1.** (Continued)

| Gene | Gene description | Pathotype[a] Species | Case | Control | Total | Odds Ratio |
|---|---|---|---|---|---|---|
| | | | N = 57 (%) | N = 40 (%) | N = 97 (%) | |
| *iucC* | Aerobactin iron transport system | ExPEC | 32 (56.1) | 21 (52.5) | 53 (54.6) | 1.2 |
| *sitA* | Iron caption system, Sit operon | ExPEC/APEC | 12 (21.1) | 8 (20) | 20 (20.6) | 1.1 |
| *terC* | Tellurium ion resistance protein | ExPEC | 5 (8.8) | 9 (22.5) | 14 (14.4) | 0.3 |
| *ompT* | Outer membrane protease gene | ExPEC/APEC | 15 (26.3) | 4 (10) | 19 (19.6) | **3.2¤** |
| *aufA* | Fimbrial subunit | ExPEC | 5 (8.8) | 2 (5) | 7 (7.2) | 1.8 |
| *yfcV* | Putative fimbriae | ExPEC | 3 (5.3) | 1 (2.5) | 4 (4.1) | 2.1 |
| *senB* | Plasmid encoded enterotoxin | STEC | 4 (7) | 3 (7) | 7 (7.2) | 0.9 |
| *tia* | Cell invasion determinant | STEC/ETEC | 16 (28.1) | 12 (30) | 28 (28.9) | 0.9 |
| *tibA* | Autotransporter and afimbrial adhesin | EPEC/ETEC | 0 (0) | 1 (2) | 1 (1) | - |
| *tibC* | TibA glycosolator | EPEC/ETEC | 0 (0) | 1 (2) | 1 (1) | - |
| *Phylogenetic group* | | | | | | |
| A | | | 19 (33.3) | 10 (25) | 29 (29.8) | 1.5 |
| B1 | | | 22 (38.6) | 17 (42.5) | 39 (40.2) | 0.9 |
| B2 | | | 1 (1.8) | 1 (2.5) | 2 (4.1) | 0.7 |
| D | | | 14 (24.6) | 9 (22.5) | 23 (21.7) | 1.1 |
| E | | | 0 (0) | 2 (5) | 2 (2.1) | - |
| F | | | 1 (1.8) | 1 (2.5) | 2 (4.1) | 0.7 |
| | | | | | | |

* 95% CI [1.6–11.2], χ2 9.2, p = 0.002. Δ 95% CI [1.1–6.5], χ2 4.98, p = 0.026. ¤95% CI [0.98–10.56], χ2 3.97, p = 0.046.P < .05 is significant. Fisher exact test was applied when the comparisons between cases and controls were <5 observations.

[a] pathotype or species most often associated with that particular gene. [b] Both EIEC and *Shigella*. EAEC, enteroaggregative *E. coli*. EIEC, enteroinvasive *E. coli*. ExPEC, extra-intestinal *E. coli*. EPEC, enteropathogenic *E. coli*. STEC, Shigatoxin producing *E. coli*. ETEC, enterotoxigenic *E. coli*. DAEC, diffusely adherent *E. coli*. APEC, avian patogenic *E. coli*.

known AAF variant in 71 (73%) of EAEC strains. The AAF/I cluster was most frequently observed in our collection (in 23%), followed by AAF/IV (22%), AAF/V (16%), AAF/III (9%), and AAF/II (8%). Importantly, we did not observe the presence of any previously unreported AAF variants.

The AAF adhesins are encoded by a four gene cluster [39], which is phylogenetically related to the usher-chaperone family of fimbriae. The AAF/I-encoding genes comprise, in order, *aggD* (chaperone), *aggC* (usher), *aggB* (minor pilin) and *aggA* (major pilin). This gene order was preserved in all AAF/I, III, IV and V clusters (Fig 1). In contrast, the published organization of AAF/II comprises two separate clusters (*aafDA* and *aafCB*), located at a significant distance on the pAA plasmid; this organization was observed for all eight AAF/II variants in the present analysis.

Nineteen strains (20%) were negative for any genes attributed to the known AAF variants; ten of those where positive for the chromosomally-encoded gene *aaiC*, and eight of these were negative for pAA-encoded genes, including *aggR*. Two *aaiC* positive- but AAF-negative strains, C251-15 and C693-09, harbored *aggR* and several *aggR*-regulated genes. Six of the nine AAF- and *aaiC*-negative strains harbored the *aggR* gene and three of those had the same virulence profile (*aap, aatA, aggR, air, capU, chuA, eilA, gad, iss, kpsE, kpsMII, ompT, sepA*) and all belonged to serotype O166:H15. Finally, the last three strains harbored a small number of *aggR*-regulated genes (*aap, capU*, ORF3, and ORF4).

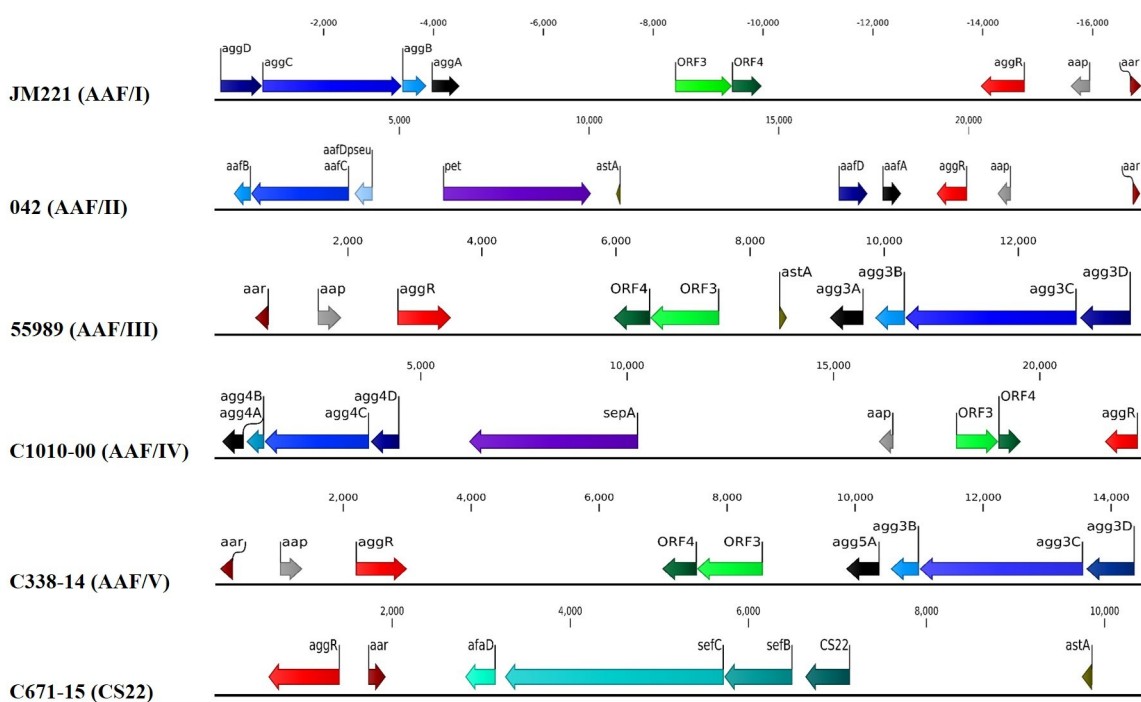

**Fig 1. Annotation of the AAF/I-V and CS22-like biogenesis gene clusters, and adjacent genes, as determined by nucleotide sequence analysis of the product amplified from prototype strains JM221, 042, 55989, C1010-00, C338-14, and C671-15.** Organization of the CS22-like gene cluster differs from AAF variants I, III-V in that the predicted minor pilin gene, *afaD*, is located adjacent to the usher-encoding gene (*sefC*) and not the major pilin gene. All ORFs encoding >50 predicted amino acids are indicated. The nucleotide positions of the predicted translational start and stop codons are also indicated. We performed phylogenetic analysis of the predicted mature major pilin proteins (the basis for assigning AAF variants). Overall, the minimum amino acid similarities of the major pilin genes within each AAF variant were as follows: AAF/I (*aggA*), 64%; AAF/II (*aafA*), 80%; AAF/III (*agg3a*), 85%; AAF/IV (*agg4a*), 80%; and AAF/V (*agg5a*), 80%. Amino acid identities of the major pilin proteins between any two major variants was less than 25% (S3 Table). Variation within *aggA* alleles identified three sub-clusters (Fig 2), designated here as subgroups a, b and c. The similarity within the three sub-clusters was high: subgroup a, 94%; subgroup b, 91%; and subgroup c, 77%.

To ascertain if the 19 AAF-negative strains manifested the defining AA phenotype, we performed the HEp-2 adherence assay. None of the strains exhibited the AA pattern. Of all the strains assayed only three strains adhered to the cells but not with an AA pattern and one strain, C682-15, was cytotoxic. This indicates that they would not be considered EAEC under the current definition.

We observed five different alleles of *aar* (here designated *aar*1-5), encoding the AggR anti-activator, with a predicted overall protein identity of 88%. These Aar alleles segregated according to specific AAF variants; e.g. Aar5 was only found among strains harboring AAF/II (S2 Table). Notably, all AAF/IV-producing strains lacked *aar*.

## A new gene-cluster in EAEC

Given the importance of adhesion in EAEC pathogenesis, we examined more closely strains negative for the five known AAF variants, but harboring the AggR regulator and other pAA-borne genes. We thereby identified three strains (C719-09, C287-15, and C671-15) whose genomes harbored a similar non-AAF fimbrial biogenesis gene cluster; the genes are predicted to assemble a more distant member of the chaperone-usher adhesin family. The cluster comprised four genes: the gene encoding the non-fimbrial ETEC colonization factor (CF) pilin

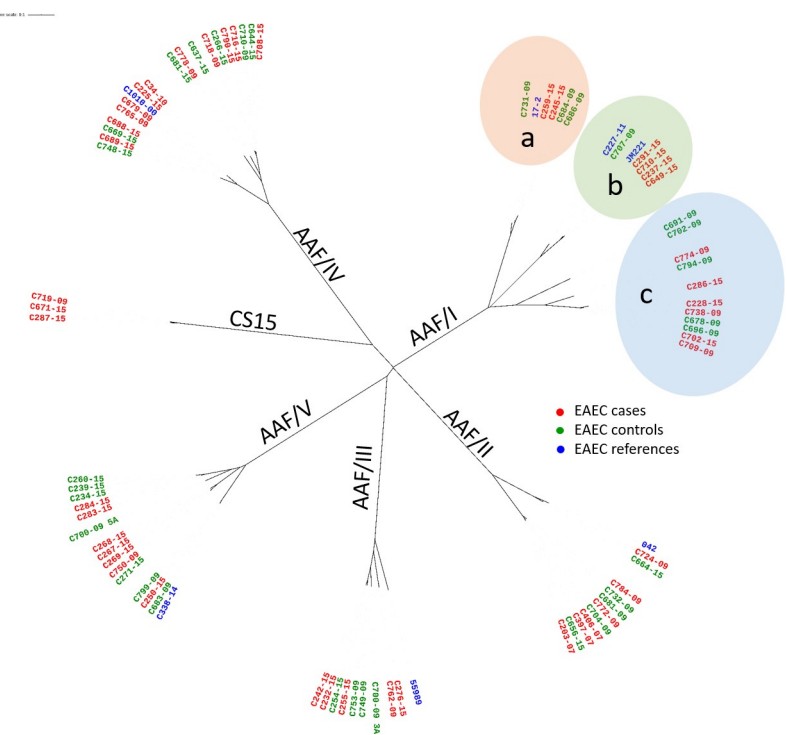

**Fig 2. Neighbor-joining tree analysis of the predicted pilin protein of AAF/I-V and CS22.** Pilin-encoding gene sequences were extracted from the EAEC isolates, translated, and truncated at the sites of proven or predicted signal peptidase cleavage. The three distinct subgroups of AggA alleles, designated as subgroups a, b and c, are shown. Derived amino acid sequences were aligned using ClustalW and an unrooted phylogram including weighted branch lengths was plotted using iTOL software.

protein called CS22 [40], predicted fimbrial chaperone and usher genes *sefB* and *sefC* originally described in *Salmonella enterica* serovar Enteritidis gene cluster *sefABCDE* [41,42], and a minor predicted fimbrial subunit annotated as "AfaD-like" in GenBank (NCBI Reference Sequence: WP_096960972.1). The organization of the CS22-like gene cluster, as determined by nucleotide sequence analysis of the product amplified from strain C671-15, is shown in Fig 1. Organization of the CS22-like gene cluster differs from AAF variants I, III-V in that the predicted minor pilin gene, *afaD*, is located adjacent to the usher-encoding gene (*sefC*) and not the major pilin gene. When comparing amino acids from the predicted CS22-like protein individually to other members of the Dr family of adhesins as well as to the AAF pilins, the predicted pilin displayed the following closest percent similarities: 24% with AfaE-8 and M-agglutinin; 23% with BmaE and AfaE-V; 22% with AfaE-I; and 20% with DraE. The similarity with the AAF archetype pilins were at most 14% (S3 Table). We have previously noted that AAF pilins exhibit an unusually high predicted pI (>8.0) [14]; the CS22-encoding gene displayed a low predicted pI of 6.0. Therefore, the CS22-like pilin cannot be considered a novel AAF pilin.

To yield additional insights into these strains, we sequenced and annotated the entire 143 kb pAA plasmid from the CS22-like containing strain C671-15 (pAA$^{C671-15}$) by using a Min-ION Mk1B (ONT) (S1 Fig). Beside the CS22-like fimbrial cluster, the pAA$^{C671-15}$ plasmid harbored several genes reported as putative EAEC virulence factors: *aar*, *aggR*, *aatABCDP*, *aap*, *sepA*, and two copies of *astA*.

To establish the prevalence of the CS22-like gene cluster among EAEC strains, we screened 469 *aggR* positive strain genomes available in EnteroBase [43], and identified 25 strains (~ 5%)

positive for EAEC CS22-like gene cluster when we employed an aa 85% identity threshold and 60% minimum alignment length (S3 Table). The alignment of the CS22-like peptide from all such strains revealed an aa identity of 83.74%-100%. The virulence gene combinations and serotypes of C719-09, C287-15, and C671-15 could be found among the 25 strains from EnteroBase (*i.e.*, O39:H49, O9:H12; *aaiC, aap, aar, aatA, aggR, astA, lpfA, pic, iss, sepA*. Notably, all CS22-like positive strains were positive for *astA, sepA, aggR*, and *aar*, the latter two being 100% identical on the nucleotide level. These data suggest that strains harboring the CS22-like adhesin are characteristically typical EAEC by genomic criteria.

Importantly, the CS22-like strains were negative for adherence to HEp-2 cells (see above). To ascertain if these strains were able to adhere to the human intestinal epithelium, we incubated CS22-like archetype strain, C671-15, with human colonic organoids and compared their ability to adhere with that of EAEC strain 042. As shown in Fig 3, adherence between the two stains was nearly equivalent, and both adhered significantly more than an *aafA* null mutant [44] that exhibits adherence defects in this model system [45,46].

## Prevalence and distribution of putative virulence genes

Seeing EAEC is highly heterogeneous with regard to putative EAEC virulence factors, we expanded our genomic search to include virulence genes from DEC and ExPEC. Among our 97 genomes, we noted the presence of a substantial number of genes characteristically associated with other *E. coli* pathotypes, such as STEC, ETEC and extraintestinal pathogenic *E. coli* (ExPEC) (Table 1). ExPEC genes found among the EAEC strains included i) yersiniabactin receptor-encoding gene *fyuA* (58%); ii) increased serum survival-encoding gene, *iss* (38%); iii) aerobactin iron transport system-encoding gene, *iucC*; iv) translocate group II capsular polysaccharide-encoding genes *kpsMII* and *kpsE* (both 36%); v) haem- utilization-encoding gene *chuA* (27%); vi) iron caption system-encoding gene, *sitA* (20%); vii) heat-resistant agglutinin-encoding gene, *hra* and viii) the outer membrane protease-encoding gene *ompT* (both 19%); ix) putative fimbria-encoding gene, *yfcV* (4%); and x) afimbrial adhesion regulator-encoding gene, *afaA-III* (2.1%). The STEC and ETEC virulence genes also found in the EAEC isolates were i) long polar fimbriae *lpfA* (48.5%); ii) cell invasion determinant *tia* (29%); iii) *irgA* homologue adhesin *iha* (27%); and iv) microcin, *mchB* (23.7%). LpfA was the only virulence factor to differ in frequency between the phylogenetic groups, as it was found in all B1 isolates, Phylogroup D (only lineage 2, see Fig 4), and phylogroup F. Seven strains (7%), including archetype strain 042, harbored the ExPEC fimbrial *aufA* gene, and these strains were also only found in Phylogroup D lineage 2.

Of the five genes encoding the SPATEs most commonly found in EAEC, the most frequent among our strains were *pic* (at 36.1%) and *sepA* (35%). The least common SPATEs were *pet* and *sigA* (both 8%)(Table 1). There was general concordance of SPATE genes with AAF variants; e.g. *sepA* was characteristically found among strains that harbored AAF/IV, and of the 34 *sepA*-positive strains, 20 harbored AAF/IV.

## Serotypes and antimicrobial characteristics

Several serotypes are considered to be markers of pathogenic *E. coli* and therefore are routinely screened for in public health and food industry settings. To ascertain the relationship between virulence factors and serotype, we applied both conventional and *in silico* serotyping. The 97 EAEC strains belonged to a diverse range and combination of O:H and phylogenetic types (S2 Table). Some O:H combinations were common, such as O153:H30 (seven strains), O166:H15 (six strains) and O99:H33 (six strains). The most common sequence type (ST) was ST10 (ten strains), followed by ST38 (seven strains), ST349 (six strains), and ST484, ST678, and ST841

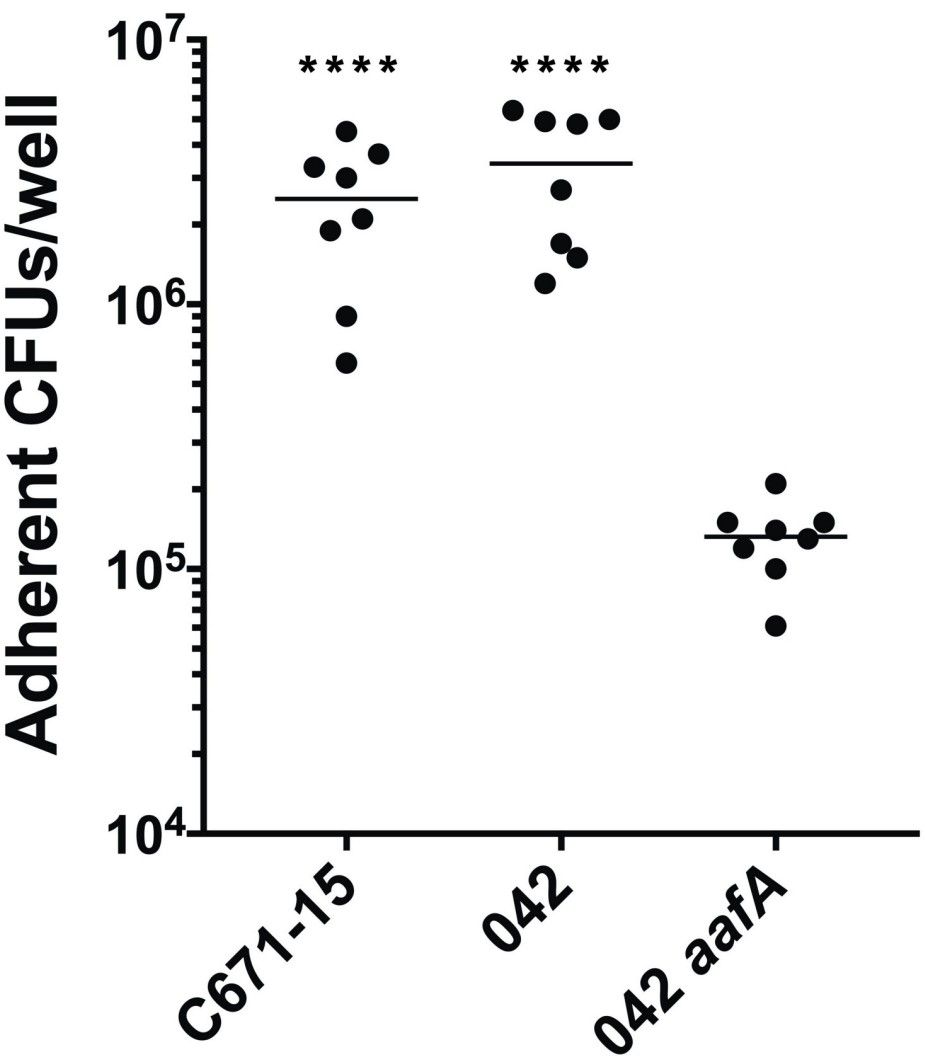

**Fig 3. Adherence to colonoid monolayers.** Colonoids monolayers were incubated with EAEC strains for 3 hours, and then adherent bacteria were enumerated. One-way ANOVA was performed on log-transformed data followed by Tukey's test for multiple comparisons. Adherence by strains 042 and C671-15 was not significantly different, but both strains adhered more than 042 *aafA* (**** p≤ 0.0001). Data shown are from two independent experiments.

(four strains each). Strains with the same serotype and ST typically harbored the same set of virulence factors. Notably, O166:H15 (ST 349) has been found in EAEC bloodstream isolates from children in Mozambique [47], and was previously implicated in a diarrheal outbreak in Japan [48].

The most commonly found H types were H30 and H4 (10 strains each), H12, H15, H27, H16 (seven strains each) and H33 (six strains). Half of the H4 strains belonged to phylogenetic group B1 and ST678, which forms a clonal group including the EAEC outbreak strain O104:H4 (Fig 4).

The majority of isolates (~80%) harbored genes conferring resistance to three or more groups of antibiotics (S2 Table). Six EAEC isolates harbored the common *blaCTX-M-15*gene [49].

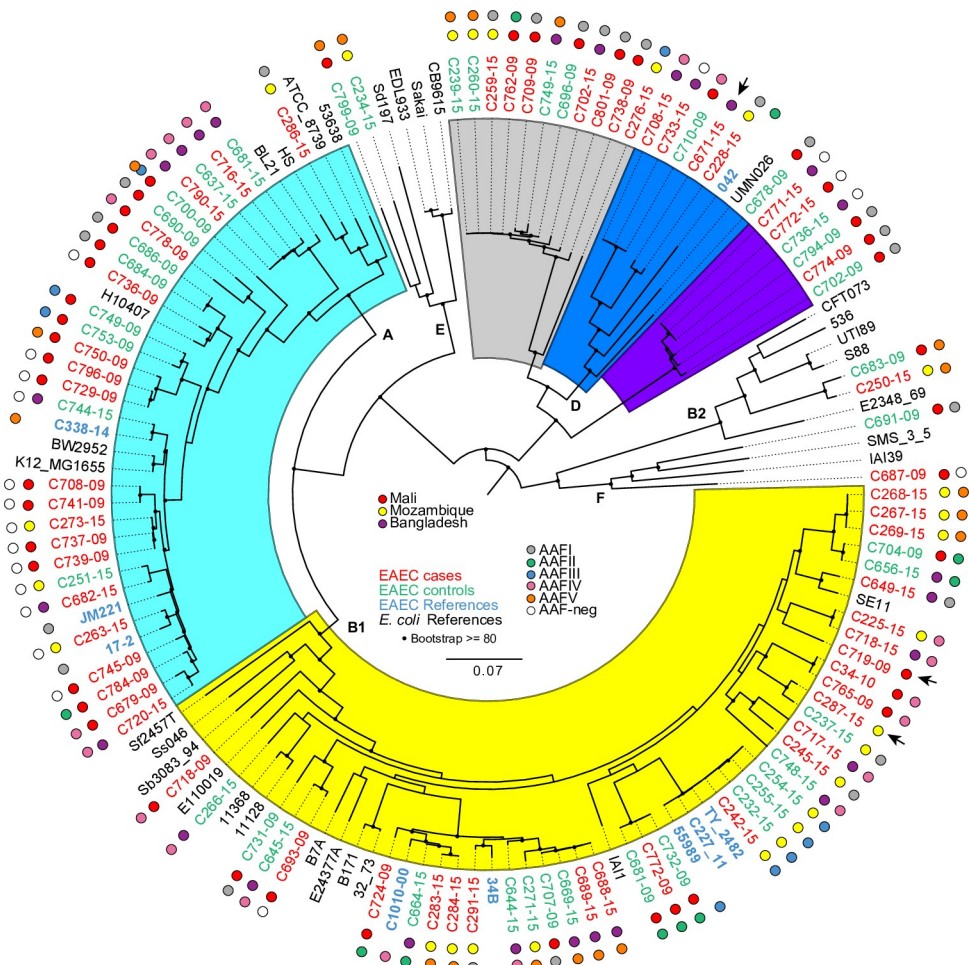

**Fig 4. Phylogenomic analysis of the EAEC genomes.** The whole-genome phylogeny was constructed from 234,371 conserved SNP sites per genome that were identified compared against the reference genome *E. coli* strain IAI39 (GenBank accession no. NC_011750.1). The color of the isolate name indicates if it is a case (red), control (green), reference EAEC (blue) or reference *E.coli* (black). The phylogeny is decorated further with the geographic origin of the isolates by the colored balls outside of each isolate label. The isolates from Mali have a red ball, Mozambique, a yellow ball and Bangladesh, a purple ball. The outermost balls for labeling indicate the presence of the AAF fimbriae. The colors for each of these genomic regions are included in the inset legend. Three isolates are indicated as containing a CS22-like gene cluster, originally described in ETEC isolates [40] by an arrow in the outer ring of labels. The *E. coli* phylogroups are designated by letters (A, B1, B2, D, E, and F) on the interior of the phylogeny based on the inclusion of the reference isolates, as well as colored sections within the figure (Phylogroup A–aqua; Phylogroup B1 yellow; Phylogroup D is separated into three sub-clusters of EAEC isolates (grey, blue and violet), with only the blue sub-cluster containing reference isolates. The tree scale indicates the distance of 0.1 nucleotide changes per site. Bootstraps values ≥80 are indicated by black circles.

## Phylogenomic analysis of the EAEC genomes

Generally, *E. coli* can be divided into seven phylogenetic groups (A, B1, B2, C, D, E, F) [50]. Forty-two percent (39 strains) of the sequenced EAEC strains belonged to the phylogroup B1, followed in frequency by phylogroup A (29 strains) and phylogroup D (23 strains) (S1 Table and S2 Table). Strains belonging to group A harbored the majority of the AAF-negative strains (13/19 strains) (Fig 4). Among the phylogroup D strains, there were three clearly defined subgroups, designated lineages 1–3. In lineage 1, all strains possessed the O166:H15 serotype and ST349 (purple); lineage 2 strains included serogroups O44 (including type strain 042), O73

and O17/O77, in combination with H18 or H34 and belonging to ST130 or NT (blue). Among lineage 3, the majority (five strains) of strains exhibited the O153:H30 serotype and ST 38 (grey).

## Genomic epidemiology of EAEC in cases of MSD

We hypothesized that EAEC isolated from cases of MSD would harbour virulence factors distinct from, or more commonly than, strains isolated from asymptomatic controls. We assessed the correlation of all AggR-dependent and SPATE-encoding genes with clinical state (Table 1) by applying Fisher's exact test. Of all genes scored, *sepA* yielded the strongest association with MSD: 27 *sepA*-positive strains originated from MSD cases and seven from controls, yielding an OR of 4.2 ($P$ = 0.002) (Table 1). When assessing the correlation with MSD for all other genes assayed, we found that the increased serum survival gene, *iss*, and the outer membrane protease gene, *ompT*, were each significantly associated with MSD (p = 0.026 and p = 0.046, respectively) (Table 1); Iss has been recognized for its role in ExPEC virulence [51–53]. We observed six distinct alleles of the *iss* gene, and three of those variants (found in seven cases) had a short signal peptide, similar to the phage λ gene *bor* [54]. The similarity between the predicted proteins from *iss* and *bor* was 92.59%. Overall, we identified 105 features that were statistically (FDR p value < 0.05) associated with symptomatic EAEC (S4 Table) and 225 features that were statistically associated with the asymptomatic EAEC (S4 Table).

## Discussion

Since the initial description of EAEC, a large number of putative virulence EAEC factors have been described. However, EAEC strains have manifested striking mosaicism and marked genomic heterogeneity, even among strains from a single epidemiologic site. Moreover, the proliferation of potential virulence factors has not yielded an improved definition of this emerging pathotype.

The most important advance in understanding the genomics and pathogenicity of EAEC has been the description of the AggR regulon. Strains harboring AggR have been termed typical EAEC [26], and most studies employing molecular diagnostics have targeted such strains. Given the challenges to performing the HEp-2 adherence assay in clinical laboratories, detection of various AggR-associated genes has served as the *de facto* standard for EAEC epidemiology, despite the lack of formal analysis substantiating this approach.

In this study, we generated the most definitive characterization to date of the pan-genome of EAEC by performing WGS on a diverse collection of strains isolated from different geographic settings, and compared these genomes with already available genomes from strains isolated from around the world. All strains harbored at least one gene shown to be under the control of AggR; the strains are thus representative of isolates that are currently considered to be EAEC in clinical and epidemiologic studies (and are typical EAEC in the formal definition).

We report here that among the isolates analyzed, all strains meeting the current definition of EAEC (adherence to HEp-2 epithelial cells) possessed one of the five reported AAF variants; strains lacking an identifiable AAF, including those harboring the CS22-like cluster, did not yield the aggregative pattern, even if they harbored AggR or other AggR-associated genes. These data suggest for the first time a molecular definition of EAEC: we propose to define EAEC as *E. coli* strains harboring AggR and a complete cluster of AAF-encoding genes (usher, chaperone, and both major and minor pilin subunit genes) *or* CS22. By applying this new definition to our strain collection, 76% (74 strains) fit the criteria (S2 Table); the other 24% characteristically harbored only a small subset of putative virulence-associated genes. Although some strains not meeting this definition may be human pathogens, we believe that they will display

different clinical, epidemiologic and pathogenetic characteristics compared with true EAEC, and therefore, they should be excluded from the EAEC appellation. Whether such non-EAEC strains will ultimately merit assignment to a new pathotype will depend on future data that suggest a definitive association with human disease.

EAEC in general was not associated with MSD in the GEMS study. To profile the virulence genes of our strain set, we blasted our 97 EAEC genomes for ~200 virulence and putative virulence genes found in DEC and ExPEC. As previously reported we found our EAEC strains to be highly diverse. Therefore, we quantified the strength of the association between cases and control calculating the odds ratio (OR). Few of the virulence genes described in Table 1 were themselves associated epidemiologically with disease. The strongest association was for the SepA protease, which has previously been associated with disease epidemiologically [7]. Interestingly, we also found an association of the *iss* gene and *ompT*. Both genes play a role in the ability of the bacterium to resist the killing effects of the host immune system: OmpT is responsible for protamine inactivation [55] and the *iss* gene confers complement resistance [56]. *iss* has been found to be associated with *E. coli* isolated from female patients with bacteremia of urinary tract origin [51] and it has been suggested that OmpT plays a multifaceted role in the pathogenesis of urinary tract infections contributing to urovirulence [57,58]. Notably, 15.5% (15 strains) of the EAEC strains harbored both the *iss* and the *ompT* genes, and this pattern was strongly associated with MSD (p = 0.017). Thirteen of these 15 dual-positive strains fit our proposed modified definition of EAEC; eleven of these strains belonged to uncommon serotypes such as O154:H21 (ST 40) and O7:H4 (ST 484) suggesting the existence of pathogenic clones. Notably, our strains belonged to 43 different ST types, including ST10, ST38, ST40, ST131, ST678, which have all been described as either statistically associated with EAEC disease or implicated in EAEC outbreaks [4,59–61].

Consistent with the implication of *iss* and *ompT*, we report that EAEC strains harbor additional putative virulence genes previously reported in ExPEC and avian pathogenic *E. coli* (APEC), and indeed EAEC strains have been implicated in urinary tract infection (UTI) [60] and bacteremia in children under five years of age [47]. A recent study reported that EAEC with ExPEC markers were distributed into three distinct phylogenetic branches [62]. The authors of this report suggested that phylogenetic group A EAEC with ExPEC markers are potential agents of UTI; in our analysis, however, we observed ExPEC markers across all phylogenetic groups. It is possible that both authentic enteric and urinary/systemic pathogens can be found among strains meeting the definition of EAEC.

The natural history of EAEC can be illuminated by global genomic analysis, whereas EAEC vaccine development will require a detailed understanding of antigenic diversity. Both areas of discussion have been advanced by our analysis. Most importantly, our data suggest that AAF adhesins may be promising immunogens based on their high prevalence and generally high amino acid conservation. As seen for other pathotypes, the most common EAEC serotypes featured characteristic combinations of putative virulence-associated genes. For example, all AAF/IV-producing strains lacked the *aar* gene (which is associated with down-regulation of the AggR regulon), and such strains commonly harbored the SepA-encoding gene. Such clones may be more diarrheagenic than other EAEC. Despite different geographic origins, all AAF/II-producing strains harbored a discontinuous fimbrial biogenesis genetic organization, as first described for the Peruvian archetype strain 042 [63].

EAEC remains a common bacterium associated with diarrhea and linear growth faltering among children in developing countries, and increasingly with extra-intestinal disease. Our analyses may aid epidemiologic investigation, vaccine development, and clinical characterization of this pathogen.

## Supporting information

**S1 Fig. Annotation of the 143,612 bp pAA plasmid from Strain C671-15 with a MinION Mk1B (ONT).** The plasmid was annotated using Rast as well as manual BLAST searches. Known virulence and putative virulence genes are shown in red. Other genes and open reading frames are shown in blue.
(TIF)

**S1 Table. List of *E. coli* and *Shigella* reference genomes included in the phylogenomic analysis of the EAEC genomes.**
(XLSX)

**S2 Table. Genomic characteristics and description of the 97 EAEC strains and six references EAEC strains including; isolate number, country, type/ condition, pathotype, phylogroup, AAF/ adhesin, putative virulence genes, serotype, fimH, Aar variant, MLST ST (CC), antibiotic resistance genes, accession number, and number of contigs.**
(XLSX)

**S3 Table. Comparisons of amino acids from the predicted CS22-like protein individually to other members of the Dr-family of adhesins as well as to the AAF pilins.**
(XLSX)

**S4 Table. Large-scale BLAST score ratio (LS-BSR) analysis of the 97 EAEC genomes: identifying 105 features that were statistically (FDR p value < 0.05) associated with symptomatic EAEC and 225 features that were statistically associated with the asymptomatic EAEC.**
(XLSX)

## Acknowledgments

We thank Susanne Jespersen Statens Serum Institut for laboratory support. We also thank Dr. Jorge Giron, University of Virginia, School of Medicine for performing the Hep2 cell adherence assay.

## Author Contributions

**Conceptualization:** Nadia Boisen, Araceli E. Santiago, Inacio Mandomando, David A. Rasko, Flemming Scheutz, James P. Nataro.

**Data curation:** Nadia Boisen, Araceli E. Santiago, Inacio Mandomando, Alejandro Cravioto, Marie A. Chattaway, Laura A. Gonyar, Søren Overballe-Petersen, Flemming Scheutz, James P. Nataro.

**Formal analysis:** Nadia Boisen, Mark T. Østerlund, Katrine G. Joensen, Araceli E. Santiago, Inacio Mandomando, Marie A. Chattaway, Søren Overballe-Petersen, O. Colin Stine, David A. Rasko, Flemming Scheutz, James P. Nataro.

**Funding acquisition:** Nadia Boisen, James P. Nataro.

**Investigation:** Nadia Boisen, Inacio Mandomando, Alejandro Cravioto, Marie A. Chattaway, O. Colin Stine, David A. Rasko, Flemming Scheutz, James P. Nataro.

**Methodology:** Nadia Boisen, Mark T. Østerlund, Katrine G. Joensen, Araceli E. Santiago, Laura A. Gonyar, Søren Overballe-Petersen, O. Colin Stine, David A. Rasko, Flemming Scheutz, James P. Nataro.

**Project administration:** Nadia Boisen, James P. Nataro.

**Resources:** Nadia Boisen, Flemming Scheutz, James P. Nataro.

**Validation:** Nadia Boisen, Søren Overballe-Petersen, David A. Rasko, Flemming Scheutz, James P. Nataro.

**Visualization:** Nadia Boisen, Araceli E. Santiago, Laura A. Gonyar, Søren Overballe-Petersen, David A. Rasko, Flemming Scheutz, James P. Nataro.

**Writing – original draft:** Nadia Boisen, David A. Rasko, Flemming Scheutz, James P. Nataro.

**Writing – review & editing:** Nadia Boisen, Mark T. Østerlund, Katrine G. Joensen, Araceli E. Santiago, Inacio Mandomando, Alejandro Cravioto, Marie A. Chattaway, Laura A. Gonyar, Søren Overballe-Petersen, O. Colin Stine, David A. Rasko, Flemming Scheutz, James P. Nataro.

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
