## [Decision Letter · Decision Letter 0]

6 Jul 2020

Dear Dr. Boisen,

Thank you very much for submitting your manuscript "Redefining enteroaggregative Escherichia coli (EAEC): Genomic Characterization of Epidemiological EAEC Strains" for consideration at PLOS Neglected Tropical Diseases. As with all papers reviewed by the journal, your manuscript was reviewed by members of the editorial board and by several independent reviewers. The reviewers appreciated the attention to an important topic. Based on the reviews, we are likely to accept this manuscript for publication, providing that you modify the manuscript according to the review recommendations. 

Sincerely,

Claire Jenkins

Guest Editor

Ana LTO Nascimento

Deputy Editor

Reviewer's Responses to Questions

**Key Review Criteria Required for Acceptance?**

**Methods**

-Are the objectives of the study clearly articulated with a clear testable hypothesis stated?

-Is the study design appropriate to address the stated objectives?

-Is the population clearly described and appropriate for the hypothesis being tested?

-Is the sample size sufficient to ensure adequate power to address the hypothesis being tested?

-Were correct statistical analysis used to support conclusions?

-Are there concerns about ethical or regulatory requirements being met?

Reviewer #1: Methods are appropriate and well described for the most part.

Reviewer #2: The objectives and the end outcome of the study needs to be better articulated. The authors have substantial work done but the objective for each section needs to be stated clearly.

Reviewer #3: Yes to all the questions. 

See the attachment for other suggestions.

**Results**

-Does the analysis presented match the analysis plan?

-Are the results clearly and completely presented?

-Are the figures (Tables, Images) of sufficient quality for clarity?

Reviewer #1: The results are clearly and transparently presented and weaknesses appropriately identified.

The supplementary tables did not convert well into PDF, and as a result, the final PDF file has what appears to be 100s of pages of empty space.

Reviewer #2: Explain and define what is a control strain and how is it different from case strain? 

The results can be divided into sections and subsections each explaining a group of idea. Introduce in a line or two why what is done in the results sections so that it is each to follow the process. 

Table 1: The legend needs to describe the method used and what is been shown in the table. 

Table 1: In the text please describe the odds ratio and why is it imp.

Reviewer #3: Yes to all the questions. 

The results of the assay of adherence of colonoid monolayers are not properly clear.

See the attachment for other suggestions.

**Conclusions**

-Are the conclusions supported by the data presented?

-Are the limitations of analysis clearly described?

-Do the authors discuss how these data can be helpful to advance our understanding of the topic under study?

-Is public health relevance addressed?

Reviewer #1: The conclusions are well supported, although the recommendation for reclassification of EAEC based on this paper may require further supportive data.

Reviewer #2: (No Response)

Reviewer #3: Yes to all the questions. 

See the attachment for other suggestions.

**Editorial and Data Presentation Modifications?**

Reviewer #1: In figure 4, the color shading of the phylogenetic groups themselves is not explained in the legend. I am assuming the colors were put in just to provide visual clarity about the separation of the groups, but given that the rest of the color schemes were clearly described, this became a bit confusing.

In Figure 3, the statistics should be shown on the figure itself.

Reviewer #2: (No Response)

Reviewer #3: (No Response)

**Summary and General Comments**

Reviewer #1: This manuscript by Dr. Nadia Boisen et al describes a thorough genomic characterization of E. coli isolates characterized as EAEC based on molecular criteria from the MAL-ED study. This is an important work in the field, as heterogeneity of EAEC has complicated clinical studies for years. The most interesting finding is that of a previously unrecognized cluster of EAEC-like organisms that express the CS22 colonization factor associated with ETEC instead of AAF adhesins. The authors propose that the current molecular characterization of EAEC be expanded to include these strains, despite their lack of aggregative adherence on HEp-2 cells, based in part on their identification in children with diarrhea, and their adherence to human colonoids in vitro.

The work as a whole is very thorough and transparently presented. The proposed reclassification is likely to engender good discussion among experts in the field, and might not be fully justified based on this current study, without further characterization in vitro and further work to associate these CS22 strains with disease (which can likely be done in the future in retrospective studies using isolates and clinical data from other longitudinal cohorts).

To improve the manuscript, I would suggest the following revisions:

1. In tables 1 and S4, there does not appear to have been a multiple comparisons correction used. I would recommend a statistical review to help determine whether this is needed in this case.

2. Table S4 lacks labeling of the columns to make the interpretation clear.

3. The “negative” Hep2 assay (lines 197-199) does not describe whether these isolates demonstrated any mannose-resistant adherence at all, or just not an aggregative pattern. Based on the discussion later on, it appears that there was no adherence at all, but this should be stated in the results section.

4. In the colonoid assays, I would like to know how other WT strains (e.g. HS and other nonpathogenic commensals) adhere in comparison to the three isolates shown in figure 3; i.e. is this specific to pathogenic strains? If not, then colonoid adherence may not be enough to put CD22 strains into the same pathogenic class as Hep-2-adherent strains. I’m also a bit surprised the authors didn’t cite the 2018 mBio paper (PMID: 29463660) that showed donor- and segment-specific differences in adherence to intestinal organoids.

Reviewer #2: A general schematic or figure summarizing the goal/objective and workflow would help in better comprehension of the paper.

Reviewer #3: (No Response)

PLOS authors have the option to publish the peer review history of their article (what does this mean?). If published, this will include your full peer review and any attached files.

Reviewer #1: Yes: Theodore S. Steiner

Reviewer #2: No

Reviewer #3: No
---

## [Editor Report · Decision Letter 1]

20 Jul 2020

Dear Dr. Boisen,

We are pleased to inform you that your manuscript 'Redefining enteroaggregative Escherichia coli (EAEC): Genomic Characterization of Epidemiological EAEC Strains' has been provisionally accepted for publication in PLOS Neglected Tropical Diseases.

Best regards,

Claire Jenkins

Guest Editor

Ana LTO Nascimento

Deputy Editor

---

## [Editor Report · Acceptance letter]

31 Aug 2020

Dear Dr. Boisen,

We are delighted to inform you that your manuscript, "Redefining enteroaggregative *Escherichia coli* (EAEC): Genomic Characterization of Epidemiological EAEC Strains ," has been formally accepted for publication in PLOS Neglected Tropical Diseases.

Best regards,

Shaden Kamhawi

co-Editor-in-Chief

Paul Brindley

co-Editor-in-Chief
